# QORA: ZERO-SHOT TRANSFER VIA INTERPRETABLE OBJECT-RELATIONAL MODEL LEARNING

## ABSTRACT

Although neural networks have demonstrated significant success in various reinforcement-learning tasks, even the highest-performing deep models often fail to generalize. As an alternative, object-oriented approaches offer a promising path towards better efficiency and generalization; however, they typically address narrow problem classes and require extensive domain knowledge. To overcome these limitations, we introduce *QORA*, an algorithm that constructs models expressive enough to solve a variety of domains, including those with stochastic transition functions, directly from a domain-agnostic object-based state representation. We also provide a novel benchmark suite to evaluate learners' generalization capabilities. In our test domains, QORA achieves $100\%$ predictive accuracy using almost four orders of magnitude fewer observations than a neural-network baseline, demonstrates zero-shot transfer to modified environments, and adapts rapidly when applied to tasks involving previously unseen object interactions. Finally, we give examples of QORA's learned rules, showing them to be easily interpretable.

## 1 INTRODUCTION

Reinforcement learning, one of the primary branches of machine learning, encompasses problems where an agent, situated in some environment, makes decisions to maximize a reward signal (Kaelbling et al., 1996; Glanois et al., 2021).This makes the field uniquely applicable to real-world tasks such as robotic manipulation (Nagabandi et al., 2020), autonomous driving (Kiran et al., 2022), and plasma confinement for nuclear fusion (Degrave et al., 2022). There is hope that the study of reinforcement learning may even lead to human-level general intelligence, but current methods are far from achieving this goal in at least three aspects. First, humans *generalize* well, transferring knowledge quickly to new settings, which allows learning to compound over time. Machine-learning methods, on the other hand, often fail when tested in scenarios that differ only slightly from how they were trained (Kansky et al., 2017). Second, current algorithms do not produce models that are *interpretable* to humans (Ghorbani et al., 2019; Glanois et al., 2021). Exchanging information is one of the most important features of social interaction, increasing trust and making knowledge acquisition more efficient; models that are difficult to understand cannot help with either of these goals. Third, humans learn *efficiently*, with many being able to confidently accomplish new tasks within hours or even minutes, while state-of-the-art reinforcement-learning agents can require decades worth of training experience (Badia et al., 2020). We refer to the set of these three properties as *GIE* (generalization, interpretability, efficiency) and propose the pursuit of all three as a path towards developing human-level intelligent systems. As a first step in this direction, we study algorithms that learn to understand their environments through interaction and observation.

More formally, the task we are interested in is *object-oriented transition modeling without domain knowledge*, which we describe here. Object-oriented representations work with entities and their attributes (e.g., position, color) rather than the tensors commonly used in deep learning (Diuk et al., 2008). Investigations in both machine learning (Chang et al., 2016) and human psychology (Spelke, 1990) have shown that understanding the world through objects and their relationships is an essential part of intelligence. Transition modeling is the process of learning predictive models of an environment's behavior; studies have shown that this is one of the primary features that enables humans to generalize (Hamrick et al., 2011) and that it can improve knowledge transfer in reinforcement learning as well (Young et al., 2022). Domain knowledge is additional information given to the agent beyond what it observes through its interactions (e.g., a list of domain-specific preconditions (Diuk

et al., 2008; Marom & Rosman, 2018)). While it is useful in certain applications, this information can be difficult or impossible to acquire; therefore, algorithms that operate without requiring such information are much more widely applicable. Although the combination of these constraints yields a challenging problem, investigating the topic is a principled path towards better techniques that possess GIE.

Unfortunately, domain-agnostic learning of object-oriented causal models is not well-studied; prior developments in reinforcement learning typically lack one or more of the three requirements stated in the preceding paragraph. Popular deep-learning approaches such as DQN (Mnih et al., 2015), Relational A2C (Zambaldi et al., 2019), Agent57 (Badia et al., 2020), and MuZero (Schrittwieser et al., 2020) do not explicitly interact with objects. The GNN architecture introduced in (Sancaktar et al., 2022) is object-oriented, but requires domain knowledge. Another line of research has studied non-deep-learning approaches to reinforcement learning, though these methods have their own limitations. Džeroski et al. (2001) introduced relational reinforcement learning, which uses first-order logical decision trees (Blockeel & De Raedt, 1998; Driessens et al., 2001) to represent policy and value functions, meaning that it does not model transitions. DOORMAX (Diuk et al., 2008) and its extensions (Hershkowitz et al., 2015; Marom & Rosman, 2018) tackle object-oriented transition modeling, but they require extensive domain knowledge to function. Schema networks (Kansky et al., 2017) use a domain-specific object representation. To our knowledge, the only notable existing techniques that satisfy all of the requirements of our problem are MHDPA modules (Vaswani et al., 2017) and the NPE (Chang et al., 2016). These methods work with objects, can be used for transition modeling, and do not require domain knowledge; however, as we show later in this paper, they do not possess GIE, which calls for the investigation of alternative methods.

Towards this end, we introduce *QORA* (*Quantified Object Relation Aggregator*), a novel algorithm for learning object-oriented transition models without domain-specific knowledge. Unlike deep-learning approaches such as MHDPA and NPE, QORA extracts relational predicates from its observations and uses statistical methods to assemble these predicates into informative hypotheses. In order to determine how well QORA meets the goal of GIE, we evaluate it in several new object-oriented benchmark environments. Through these experiments, we show that QORA improves upon prior work in all three aspects. In particular, we demonstrate: a reduction in sample complexity of almost $10,000\times$ relative to neural-network methods; zero-shot transfer to more complex scenarios; continual learning of new interactions; and easily-interpretable rules based on elegant first-order logic formulas. Thus, QORA lays a promising path towards progress on this challenging problem, opening the door for future work to tackle tasks such as playing more-complex games and controlling mobile robots. To support reproduction and extension of our results, the source code of both QORA's reference implementation and our benchmark suite will be made available online.

## 2 OBJECT-ORIENTED REINFORCEMENT LEARNING

We operate in an framework similar to the OO-MDPs from Diuk et al. (2008) where each state $s$ is represented as a set of objects $\{o_i\}$. Environments consist of a tuple $(M, C, S, B, A, T)$. $M$ is the set of *member attribute types*, each of which has a name (e.g., "position") and a size (the dimensionality of its values, e.g., two for $(x, y)$ coordinates, three for RGB color). $C$ is the set of *class types*, each of which consists of a name (e.g., "player") and a set of attribute types $c.attributes \subseteq M$. Every object $o$ belongs to a class and contains attribute values corresponding to the attribute types of its class. Each attribute value is a $d$-element integer vector where $d$ is the size of the attribute's type. $S$ is the set of all valid states. $B(s)$ is the distribution of initial states, which may be parameterized to produce specific types of levels for testing. $A$ is the finite set of actions available to the agent. $T(s'|s, a)$ is the environment's transition probability distribution that gives the probability of moving to state $s'$ after taking action $a$ in state $s$. Each object is given a unique integer id, arbitrarily assigned for starting states and not modified by $T$, so that the learner can detect what changes have occurred during a transition. Environments must satisfy the Markov property (Sutton & Barto, 2018), i.e., $T(s'|s, a)$ does not depend on any past transitions or other hidden information.

Note that there is no reward signal $R$ in the environment definition. This is because we are solely concerned with creating models of the transition behavior $T$. As such, our learning algorithms receive observations $(s, a, s')$ without making decisions; instead, actions are chosen by some other process, in our case the "random agent" that samples uniformly at random from $A$ at every step

(i.e., online off-policy learning (Sutton & Barto, 2018)). We measure the accuracy of an algorithm's model $\hat{T}$ by calculating the Earth Mover's Distance (Werman et al., 1985; Rubner et al., 1998) between the estimated distribution over future states and the true distribution over future states. This allows us to consider the actual deviation between predicted states, rather than just the difference between the probability values as with other statistical distance metrics. Earth Mover's Distance does so by incorporating a ground distance metric that gives the distance between states. For this, we use $d(s_1, s_2) = |s_2 - s_1|_1$, where $s_2 - s_1$ computes the difference in each object's attribute value from $s_1$ to $s_2$ (i.e., diff) and $| \cdot |_1$ is the sum of the absolute value of each attribute value difference (i.e., summed L1 norm over all object attribute values). Our goal is to model $T$ as closely as possible, i.e., minimize the Earth Mover's Distance on arbitrary transitions.

Learning an environment's transition dynamics given *only* $(s, a, s')$ observations, with no additional domain-specific knowledge or assumptions, is a significant challenge. Since object sets have no predefined order or size, states in this formulation are not always representable as scalars or vectors as typical of prior work (Strehl et al., 2007). This precludes the use of any methods that require fixed-sized inputs. Additionally, unlike in previous work (Diuk et al., 2008; Marom & Rosman, 2018), the learner here is not given direct access to the most important information; instead, anything it needs (e.g., relational predicates) must be generated from $s$. The goal is therefore not just to learn the behavior of the environment, but also to discern what information must be extracted in order to compute this behavior efficiently. While it would hypothetically be possible to simply memorize every seen observation and compute predictions on $(s, a)$ without using the actual content of $s$, this would be problematic for several reasons. The space $S$ and therefore also $S \times A$ are extremely large: even if limiting to small ($8 \times 8$) levels in our doors domain, $|S \times A|$ is in the trillions. More importantly, this is not an efficient way of learning. An ideal algorithm would produce a model of $T$ that *generalizes*, i.e., after being trained on only a small fraction of possible transitions, it generates accurate predictions even on unseen inputs.

## 3 QORA

We now describe QORA, a novel object-oriented model-learning algorithm that generates probability distributions over predicted states. We separate transition rules based on the (class, member attribute type, action) $(c, m, a)$ triplet they apply to and predict changes independently for each attribute of every object. Each of QORA's transition rules consists of *conditions* and *effects*. Effects add a constant value to an attribute, i.e., $o_i.m\ \mathrel{+}= v$ for some member attribute $m$ and constant $v$, and are easy to calculate when observing transitions by simply subtracting each object's attributes from previous state to next state (i.e., $s' - s$). Conditions determine the information that a rule extracts from observed states. In order to solve a wide class of problems, we allow conditions to express complex first-order logic formulas. The building blocks of these formulas are predicates of two forms: single-object predicates checking equality with a constant (i.e., $P(o_1)$: $o_1.m = v$) and pairwise predicates with relative difference equal to a constant (i.e., $P(o_1, o_2)$: $o_2.m - o_1.m = v$). These predicates can be easily extracted from an observed set of objects and, most importantly, there will only be a *finite number* of true equality predicates to extract on any given step.

Finding the correct conditions to inform a rule is not as trivial as calculating effects. The amount of information contained in even a small set of objects is staggering and the set of possible first-order logic formulas (i.e., hypotheses) grows rapidly as the number of terms increases. Since we have no *a priori* limit on the complexity of an environment's conditions, enumerating all candidates in the hypothesis space is impossible. Instead, we run an iterative algorithm that begins with small hypotheses and builds up to more-complex, more-informed hypotheses. However, this isn't sufficient by itself; it turns out that the number of possible connectives and quantifiers makes it infeasible to generate combinations of even a small number of terms. To address this, we introduce a way to express rich formulas of nontrivial size without explicitly listing connectives and quantifiers. Applying these ideas to the object-oriented setting requires some additional complexity, so we first describe our iterative rule-construction algorithm in terms of discrete function learning.

### 3.1 PREDICATE FUNCTION LEARNING

Consider the problem of learning an arbitrary function $f : \{0, 1\}^\eta \rightarrow \mathbb{Z}$, where the behavior of $f$ is fully determined by some subset of $\omega \ll \eta$ elements in the input vector called the "essential

subset". In addition to discovering the actual behavior of $f$, our algorithm must discern which $\omega$ input elements (representing predicate truth values) comprise the essential subset. Note that $\omega$ is not known, so there are $2^\eta$ possible subsets to check, and $\eta$ may be arbitrarily large – potentially several thousand even in a simple domain. Rather than evaluating this entire power set, we test the predictive capacity of each predicate *individually* and combine informative predicates together to construct the essential subset. This is based on the intuition that although an environment's rules *may* have arbitrarily complex conditions, most require very little information; thus, we explore simple rules first, only increasing complexity as necessary to improve prediction accuracy. This also leads to better generalization, as the learned rules depend only on the essential information.

To implement our constructive algorithm, we first define *candidates*. These are pairs $(u, \hat{f})$ comprising a set $u \subseteq \{1, \ldots, \eta\}$ and a function approximation $\hat{f} : \{0,1\}^{\hat{\omega}} \times \mathbb{Z} \to \mathbb{R}$, where $\hat{\omega} = |u|$. The set $u$ represents the $\hat{\omega}$ elements of the $\eta$-element input vector that the candidate takes as input. We use the notation $u(x)$ to mean "the elements of input vector $x$ corresponding to the indices in set $u$". The function $\hat{f}$ is the candidate's approximation of $f$ that returns a probability estimate $\hat{f}(u(x), y) = \hat{P}(f(x) = y)$, which can also be viewed as the conditional probability $\hat{P}(y|u(x))$ of observing output $y$ when the function's input is $x$. The *best* candidate, which most accurately models $f$, is the one that maximizes the following score function:

$$\mathcal{S}(\hat{f}) = E[\hat{f}(x, f(x))] = \sum_{(x,y)} \hat{P}(y|u(x))\hat{P}(u(x), y) = \sum_x \frac{\sum_y \hat{P}^2(u(x), y)}{\hat{P}(u(x))}. \tag{1}$$

This metric takes values in $[0, 1]$ representing the candidate's expected confidence in the correct output on a randomly-sampled input. We implement the function approximators by storing the full joint probability tables $\hat{P}(u(x), y)$, incrementing counters to keep track of observations.

To organize the candidates, we maintain a *working set*, initialized with all singleton ($\hat{\omega} = 1$) candidates, a *hypothesis list*, initially empty, and a *baseline candidate*, with $\hat{\omega} = 0$, which estimates the unconditional probabilities of each observed output $\hat{P}(y)$. As observations are recorded, candidates are updated and their scores are recalculated. Once we are confident that a candidate is performing better than the baseline (e.g., by comparing confidence intervals over the score metric) , we move that candidate from the working set into the hypothesis list. This list is kept in descending order of score so that the *best hypothesis* is always at the front. The iterative construction takes place any time a candidate is moved into a hypothesis list, or whenever a new hypothesis becomes the best. Then, new candidates are created and added to the working set by pairwise combining each hypothesis with the current best (i.e., taking the union of their $u$ sets). As a form of boosting (Schapire, 1990), candidates in the working set do not receive observations that the best hypothesis predicts with high confidence, thereby magnifying the score of candidates that are well-suited to correct existing errors.

## 3.2 QUANTIFIED FUNCTION LEARNING

Though the above algorithm works well for learning functions over a given set of predicates, it cannot be directly applied to the setting of object-oriented reinforcement learning due to the complications introduced by quantifiers. Single-object predicates, e.g. $P(o_1)$: $o_1.color = 1$, are simple to deal with, but multi-object relations always involve some sort of quantifiers. For example, consider a domain in which a player moves around in a 2d grid but is blocked by walls in adjacent squares, e.g., if a wall is positioned directly to the right of the player: $P(o_1)$: $\exists o_2 \in walls : o_2.pos - o_1.pos = (1, 0)$. These predicates may be arbitrarily complex, e.g., if the player interacts with both walls and "doors" that only let through players of the same color:

$$P(o_1): (\exists o_2 \in walls : o_2.pos - o_1.pos = (1,0)) \vee$$
$$(\exists o_3 \in doors : o_3.pos - o_1.pos = (1,0) \wedge \neg(o_3.color - o_1.color = 0)). \tag{2}$$

With multiple quantified expressions containing several inner conditions combined with an arbitrary sequence of conjunctions, disjunctions, and negations, it becomes infeasible to enumerate even just the formulas generated by combining existing hypotheses. Instead, we expand the scope of each candidate to subsume these combinations by introducing *relation groups* and *compound predicates*, two structures that we use to represent and evaluate first-order logic formulas more compactly.

Relation groups express a set of quantified predicates; for example, two relation groups would be used to represent (2), one for $\exists o_2 \in walls : o_2.pos - o_1.pos = (1,0)$ and one for $\exists o_3 \in doors :$

$o_3.pos - o_1.pos = (1,0) \wedge \neg(o_3.color - o_1.color = 0)$. Rather than combining the $k$ predicates with conjunctions and disjunctions, we instead evaluate each and produce a binary string of length $k$ (e.g., for a relation group with predicates $P_1$ and $P_2$, a case that satisfies $P_1(o_1, o_2) \wedge \neg P_2(o_1, o_2)$ generates the string 10). Then, instead of evaluating a particular quantifier ($\exists$ or $\forall$), we calculate this string for each relevant object pair and track which of the $2^k$ possible strings have been detected, resulting in a combination of existential possibilities. A universally quantified rule can then be expressed by considering the state where *only* the desired binary string was tracked, while an existential rule can be expressed by considering *any* state where the desired binary string was tracked. This process is illustrated in Figure 4 in the Appendix.

A compound predicate is a set of relation groups with each group's quantifier bound to a different type of object (e.g., in (2), $P$ would be represented by a single compound predicate). The value of a compound predicate is produced by concatenating all of its member groups' values. This setup allows learning arbitrary existential *and* universal rules, with complex inner predicates, as part of a *single candidate* just by counting observations.

### 3.3 COMPLETING QORA

To put the function-learning algorithm into use in QORA, we create *effect predictors*, one for each $(c, m, a)$ (class, member attribute type, action) triplet. When an observation $(s, a, s')$ is received, we enumerate effects in $s' - s$. For each change $v$ in attribute value $m$ of an object $x$, we send a training sample $(o_i, s, v)$ to the effect predictor for $(o_i.class, m, a)$. This effect predictor extracts predicates from $s$, evaluates them, and sends both the predicate values and observed effect value to its own instance of the quantified function-learning algorithm. This process has only a single hyper-parameter, $\alpha$, which controls the confidence level of the score intervals used to compare candidates (lower $\alpha$ leads to wider intervals). To generate a prediction, QORA constructs a joint distribution from all predicted effects based on the assumption that each $(c, m, a)$ triplet is independent.

## 4 EXPERIMENTS

To evaluate QORA and compare it to previous work, we set up tests that focus solely on each method's ability to learn from data. Each algorithm is run off-policy, with actions being chosen by a random agent and observations fed sequentially (i.e., online) *without* any sort of replay mechanism (Mnih et al., 2015). When running an experiment, we generate some number of initial states and step through $n$ random actions in each to yield the total number of observations.

### 4.1 BENCHMARK ENVIRONMENTS

We describe four testing domains, ranging from simple to complex. Although QORA could be applied to environments of arbitrary complexity, these were designed to focus on particular aspects of an algorithm's learning while being easy for humans to understand. Each domain has several parameters (e.g., width, height) that control the initial states it will generate for testing. For example states, see Figure 5 in the Appendix. Note that these settings have no effect on the behavior of the environment, i.e., $T$; therefore, by changing the parameters from training to test time, we can evaluate a learner's knowledge transfer capabilities. Effective generalization is facilitated by learning *rules*, which are distinct components of the model that typically require a relatively small amount of information to make predictions (e.g., "when the move-right action is taken, the player moves to the right unless blocked by a wall").

**Walls**  The most basic testing environment, walls is a baseline for the ability to learn relational rules. The two classes of object, player and wall, both only have position attributes. There are five actions: movement in each direction (left, right, up, down) and stay. If a movement action is chosen, the player will move one unit in the intended direction unless there is a wall blocking the position it would move to; if the stay action is taken, nothing happens. Walls are never modified by any action.

Though this domain sounds simple from a human perspective, it is important to highlight the difficulty that arises when a learner has no prior knowledge of the world. If we must learn $T$ from scratch, making only weak assumptions about its form, the sheer amount of information contained in even an $8 \times 8$ world of the walls environment is immense. It is *possible*, for example, that the

player's movement is determined not by the presence of adjacent walls but by some convoluted function based on its position and the presence of some arbitrary set of walls scattered throughout the world. Though it may seem obvious to a human, the learner has no way to rule this out *a priori*. Thus, as there are almost 300 basic facts (e.g., "the player is at position $(x, y)$") that can be observed in an $8 \times 8$ level (36 player positions within the world's border, 64 wall positions, and $13^2$ possible player-wall relative positions), and the number of possible formulas is super-exponential in this number, searching through the entire space of conditions is intractible. Previous works have made strong assumptions Hershkowitz et al. (2015) or required the correct formulas to be given for each domain (Diuk et al., 2008; Marom & Rosman, 2018), significantly reducing the difficulty of their problems while also reducing the generality of their solutions.

**Lights** The lights domain is a relational test *not* involving a grid-structured world. Instead, there is a single switch object and several lights that can be either on or off. All objects have an id; two actions, increment and decrement, can be used to change the switch's id. When the toggle action is taken, any light with the same id as the switch will toggle its state.

**Doors** This domain adds significant complexity to the types of rules that the agent must learn. In this environment (an extension of the walls domain), there is an additional object class called door and an additional member attribute called color that takes values in $\{0, 1\}$. The player and doors each have a color. Doors act like walls, except that the player can step onto doors that share its color. There is also a new action called change-color that allows the player to swap its color when not standing on a door. As an example of the formalism described in Section 2, the types in the doors environment can be expressed as:

$M = \{(\text{position}, 2), (\text{color}, 1)\}$

$C = \{(\text{player}, \{\text{position}, \text{color}\}), (\text{wall}, \{\text{position}\}), (\text{door}, \{\text{position}, \text{color}\})\}$

$A = \{\text{STAY}, \text{MOVE\_LEFT}, \text{MOVE\_RIGHT}, \text{MOVE\_UP}, \text{MOVE\_DOWN}, \text{CHANGE\_COLOR}\}$

**Fish** The fish domain is designed to test an agent's ability to learn stochastic transitions. In this environment, there are walls and some number of "fish". The agent can take two actions: stay, which does nothing, and move, which causes each fish to move around randomly. Specifically, each fish will choose a direction uniformly at random from {left, right, up, down} and move one unit in that direction unless blocked by a wall. To fully solve this domain, an agent must learn the conditional probability of each movement direction based on the existence of surrounding walls; i.e., it must be able to generate a *distribution* over possible future states.

## 4.2 RESULTS ON WALLS DOMAIN

We begin by evaluating two neural-network architectures and QORA on the walls domain. Results are shown in Fig. 1. QORA is each run 100-1000 times per experiment to generate average results, but due to the computational cost, this could not be done for the neural networks; instead, after tuning hyperparameters and architectures, we report a single run with the best settings for each variant. The neural networks tested are two *object-based* architectures, which receive a list of objects similar to the representation QORA uses except that classes are one-hot encoded (QORA uses integer class IDs). NPE is an implementation of the Neural Physics Engine (Chang et al., 2016) *without* neighborhood masking to ensure it has no domain-specific information. MHDPA is an architecture using multi-head dot-product attention modules (Vaswani et al., 2017) to capture relations. Details on both can be found in the Appendix. Note that although both architectures are highly complex relative to the domain they are tested on, neither is able to achieve perfect accuracy even after two *million* observations, as shown in Figure 1a; they both saturate at around 0.5, which implies they learned how walls work (i.e., that they do not move) but not how the player works (i.e., that it interacts with walls). Also note the effect of increasing $n$, the number of steps taken per level; the networks learn much slower, and in particular, the MHDPA with $n = 100$ never drops to 0.5 error. In Figure 1b, we train the networks for two million observations in $8 \times 8$ worlds before transferring them (without further training) to $16 \times 16$ worlds. The substantial increase in error immediately after transfer indicates that the networks were using some extraneous information specific to the $8 \times 8$ levels.

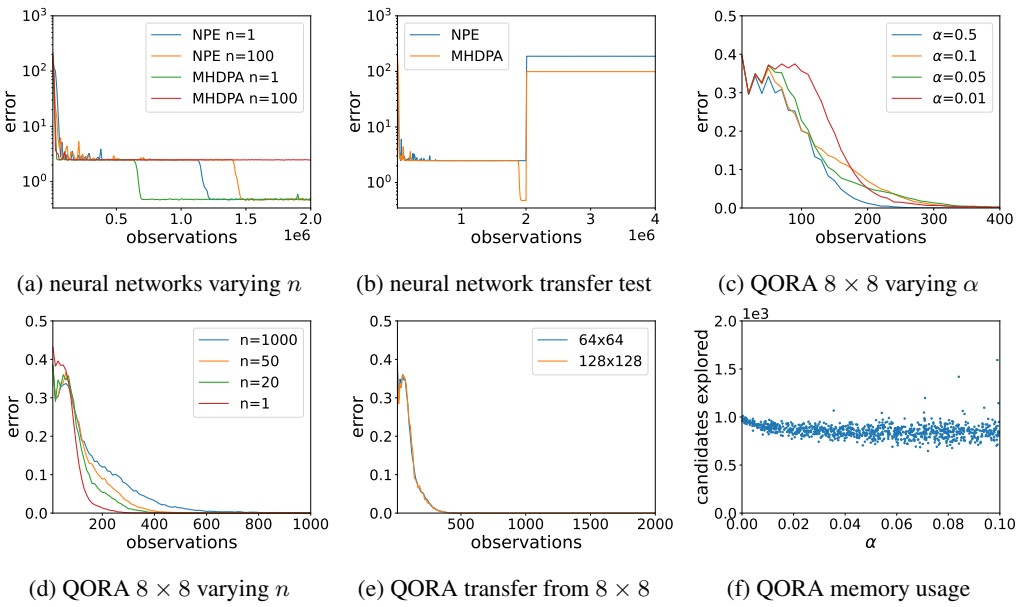

(a) neural networks varying $n$    (b) neural network transfer test    (c) QORA $8 \times 8$ varying $\alpha$

(d) QORA $8 \times 8$ varying $n$    (e) QORA transfer from $8 \times 8$    (f) QORA memory usage

Figure 1: Tests results from the walls domain. When not specified, we use $n = 20$ and $\alpha = 0.05$ for QORA. (a) Prediction error in $8 \times 8$ worlds for both deep-learning methods. (b) Transfer learning test $8 \times 8 \rightarrow 16 \times 16$ with both deep-learning methods, with no training after the transfer at two million observations. (c) QORA prediction error in $8 \times 8$ worlds, varying $\alpha$. (d) QORA prediction error in $8 \times 8$ worlds, varying $n$. (e) Zero-shot transfer to larger levels with QORA after training in $8 \times 8$ worlds for 1,000 observations. (f) QORA memory usage for varied $\alpha$ in $8 \times 8$ worlds.

In Figure 1c, we test QORA with different values of its $\alpha$ hyperparameter. On this domain, they all converge quickly to zero error (roughly 3-4 orders of magnitude faster than the neural networks reach their lowest error), though lower $\alpha$ values do lead to slightly slower learning, as expected, since the algorithm becomes more conservative with selecting hypotheses. In Figure 1d, we test QORA with varying values of $n$, showing again that lower $n$ leads to faster learning due to the presence of more variety in the training data. In all cases, QORA is still significantly more efficient than the deep-learning baselines. In Fig. 1e, we see QORA demonstrate zero-shot transfer to a new type of level layout: after being trained for 1,000 observations on $8 \times 8$ levels, the learned model is transferred without further updates to $64 \times 64$ and $128 \times 128$ environments, the latter of which contain almost five *thousand* objects. The transferred models never make any errors; this shows that in all runs of the experiment, QORA deduced the correct relational rules (i.e., checking for adjacent walls) rather than using rules specific to the levels it was trained in. Specifically, QORA learns rules including the following, where $o_1$ is a player:

$$P(o_1): \exists o_2 \in \text{walls} : o_2.pos - o_1.pos = (0, 1) \tag{3}$$

$$o_1.pos \mathrel{+}= \begin{cases} (0,0) & P(o_1) \\ (0,1) & \text{otherwise} \end{cases} \tag{4}$$

Because QORA's learning process involves iteratively constructing more candidates, it is important to analyze its memory use, which we measure by counting the total number of candidates a model explores during a run (i.e., the sum of the number of candidates ever added to the working set for each effect predictor). Note that with this scheme, many candidates are counted multiple times, since they are tracked separately for each $(c, m, a)$ triplet. In Fig. 1f, we plot the total number of candidates explored after 1,000 observations for many different runs with randomly generated $\alpha$ values in $[0, 0.1]$. Note that these are all the candidates QORA generated, *not* all candidates used: the learned rules consist of a single candidate per $(c, m, a)$ triplet, like the one previously shown. The lower bound on memory use is quite low, around 750 at every value, but the upper bound increases with $\alpha$ due to the particular random series of observations that each learner was exposed to. However, these values of $\alpha$ are quite high, shown only for completeness; most applications would

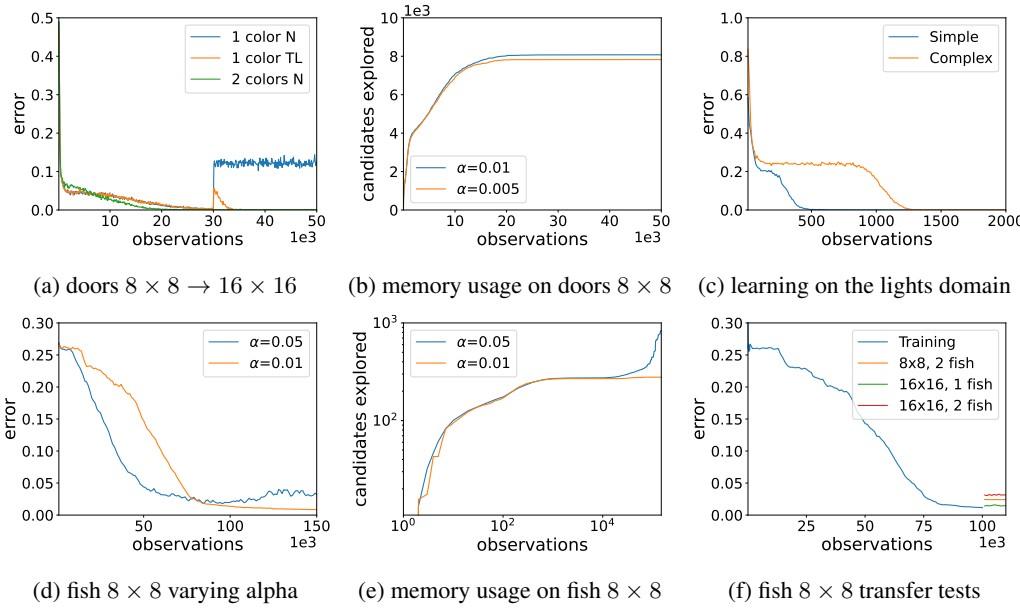

(a) doors $8 \times 8 \to 16 \times 16$    (b) memory usage on doors $8 \times 8$    (c) learning on the lights domain

(d) fish $8 \times 8$ varying alpha    (e) memory usage on fish $8 \times 8$    (f) fish $8 \times 8$ transfer tests

Figure 2: QORA on the doors, fish, and lights domains. (a) Testing continual learning with transfer from $8 \times 8$ to $16 \times 16$ levels after 30k observations, with (TL) and without (N) additional training after transfer. (b) Total number of candidates created during training (no transfer). (c) Comparing two learning modes in the lights domain: "simple" (initial training for 1,000 observations with levels with 2-10 lights, then training on levels with 2-100 lights after transfer) and "complex" (training from scratch for 2,000 iterations on levels with 2-100 lights). Since QORA is able to transfer knowledge effectively, training can begin in simpler environments to improve overall efficiency. (d) Prediction error in $8 \times 8$ worlds, varying $\alpha$. (e) Memory usage over time in $8 \times 8$ worlds. (f) Transfer tests to larger worlds with more fish, showing that QORA maintains a low error rate.

never use $\alpha > 0.05$, and generally it will be several times lower than even that. This is especially true for domains as difficult as doors and fish, which we analyze next along with the lights domain.

### 4.3 RESULTS ON LIGHTS, DOORS, AND FISH DOMAINS

We now move to the more complex domains. Because the neural methods failed on the simpler walls domain, we only test QORA here. Beginning with doors, in Figure 2a we demonstrate several variations of $8 \times 8 \to 16 \times 16$ transfer after 30,000 observations: two tests beginning in a simpler variant of the domain (with only a single door color) followed by transfer either with or without post-transfer model updates and one test that trained in full-color worlds. Similarly to what we demonstrated in Figure 1e, when trained in 2-color worlds, QORA immediately transfers to larger levels with no additional error. In the runs where QORA trains in levels with only a single door color, QORA does not immediately achieve zero error post-transfer, but this is to be expected since it is impossible to distinguish between the "correct" hypothesis (i.e., comparing player color to door color) and other "incorrect" hypotheses (e.g., simply checking the player's current color). Still, QORA maintains its knowledge of wall interactions, thereby achieving lower error post-transfer than an untrained model would. When post-transfer learning is enabled, QORA is able to quickly choose between the previously-equivalent hypotheses, re-converging to zero error within 5,000 additional observations.

Since the doors domain is significantly more complex than the walls domain (more classes, more attributes, and more complex transition rules), QORA generates a larger number of predicates while learning. In Fig. 2b, we see that QORA's predicate generation rate is dependent upon the progress it makes – QORA initially generates predicates rapidly as it learns to predict *most* interactions (i.e., learning how walls work), then slowly adds more as it eliminates the rest of its errors (i.e., learning how doors work), finally leveling off once it achieves $100\%$ predictive accuracy. This is typically

achieved with rules similar to the following:

$$P_1(o_1): \exists o_2 \in \text{walls} : o_2.pos - o_1.pos = (0,1) \tag{5}$$

$$P_2(o_1): \exists o_2 \in \text{doors} : o_2.pos - o_1.pos = (0,1) \wedge \neg(o_2.color - o_1.color = (0)) \tag{6}$$

$$o_1.pos \mathrel{+}= \begin{cases} (0,0) & P_1(o_1) \vee P_2(o_1) \\ (0,1) & \text{otherwise} \end{cases} \tag{7}$$

We now move on to the lights domain. In Figure 2c, we demonstrate a major benefit of QORA's generalization capabilities: QORA can be trained rapidly in smaller, simpler scenarios and immediately transferred to more difficult settings. When compared to training QORA from scratch in more complex instances of the domain, we see that the pre-training allows QORA to converge to perfect accuracy almost $3\times$ faster.

We finish our experiments by evaluating QORA on the stochastic fish domain. Though the rules are simple, this domain is actually the most difficult for QORA to learn. Due to the stochastic nature of the domain, the $\mathcal{S}$ score of the best candidate (i.e., the one that checks for walls around the fish) is only slightly higher than completely random guessing ($\approx 0.3$ vs. $\approx 0.2$), making it difficult to determine which predicates are actually useful. Nonetheless, as shown in Fig. 2d, QORA achieves low error within 100K observations, meaning that it is accurately predicting the entire *distribution* over possible future states.

Since the fish domain only has a single mutating action, move, predicates are not double-counted as they are in the walls domain. As shown in Fig. 2e, this results in much lower memory usage. With $\alpha = 0.01$, QORA rapidly accumulates predicates as it observes the world for the first time, then quickly levels off as new predicates no longer give better information. With $\alpha = 0.05$, QORA generally behaves similarly, but because it accepts candidates more greedily, it will *occasionally* add extraneous predicates. This leads to increased memory usage and reduced accuracy. Fortunately, simply lowering $\alpha$ makes this significantly less likely and leads to minimal rules based on the presence of walls surrounding each fish. This alpha tuning procedure is quite straightforward for any environment.

Finally, we evaluate QORA on several fish transfer tasks, shown in Fig. 2f: after training for 100K observations in $8 \times 8$ worlds with a single fish, we transfer with no additional learning to larger levels and multi-fish worlds. Though QORA loses some accuracy after transfer due to the increased complexity of these tasks, we still achieve relatively low error even on the $16 \times 16$ levels with two fish, where QORA must predict the probability of *every possible future position* of *both* fishes. Note also that the increase in error is partially due to there being more fish, since the error metric is cumulative over each object. Overall, these results demonstrate that QORA possesses powerful knowledge-transfer capabilities.

## 5 CONCLUSION

We introduced *QORA*, an algorithm capable of learning predictive models for a large class of domains by directly extracting information from object-oriented state observations. We demonstrated that QORA achieves zero-shot transfer using interpretable relational rules and is capable of rapid continual learning while simultaneously having orders-of-magnitude better sample efficiency than deep-learning approaches. This contribution opens a new path for future developments in transition modeling and the wider field of reinforcement learning; in particular, QORA's use of explicit causal rules will enable the creation of highly efficient planning and exploration algorithms for previously-inaccessible lower-level domains.

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

## A  APPENDIX

### A.1  NEURAL NETWORK BASELINE ARCHITECTURES

Our implementation of the NPE is shown in Figure 3. It is based on the description in (Chang et al., 2016) with the addition of the pre-encoding network $f$. The particular architectures of $f$, $g$, and $h$ are our own, based on several iterations of network design improvement. The total number of parameters is in this structure is 6,944. We tried more complex architectures, including other ways of encoding output (e.g., using attribute deltas like QORA), without noticeable improvement.

The MHDPA baseline consisted of a single 5-head dot-product attention layer (as implemented by PyTorch) followed by a per-object encoder similar to the $h$ module in the NPE that appends the action to each object. The total number of parameters in this structure is 2,784. We tested more complex architectures (e.g., a pre-encoder), but they only led to slower learning without improving accuracy.

For both NPE and MHDPA, we used 10 batches of 100 observations per epoch. The learning rate was 0.01 for NPE and 0.005 for MHDPA. We used stochastic gradient descent with momentum of 0.9 and L1 loss (since outputs were linear).

### A.2  RELATION GROUP VISUALIZATION

In Figure 4, we show a visualization of the process of encoding a state observation into a relation group.

### A.3  ENVIRONMENT VISUALIZATIONS

In Figure 5, we show example levels for two of our domains: walls and fish.

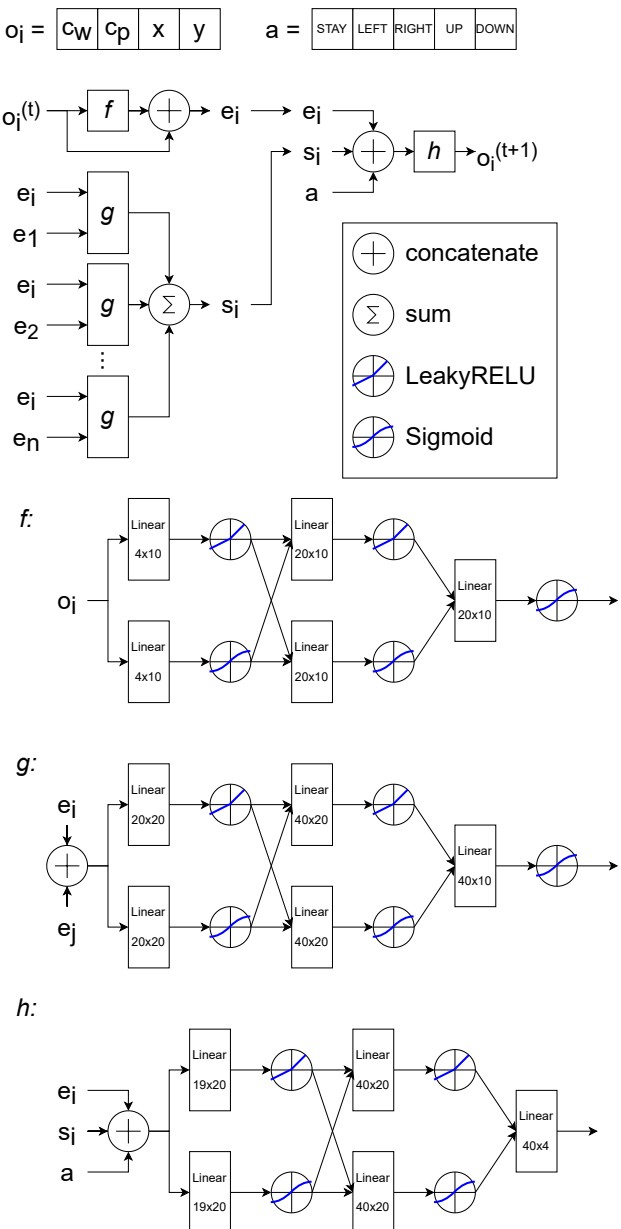

Figure 3: Architecture of our NPE baseline. Let $o_i$ be an object, which is encoded with four attributes: its one-hot class (wall or player, $c_w$ or $c_p$ respectively) and its position ($x$ and $y$ coordinates). Let $a$ be the current action, which is one-hot encoded. Objects go through a pre-encoding stage where features can be extracted ($f$) followed by a pairwise computation ($g$) and a final output calculation stage ($h$). More details can be found in Chang et al. (2016).

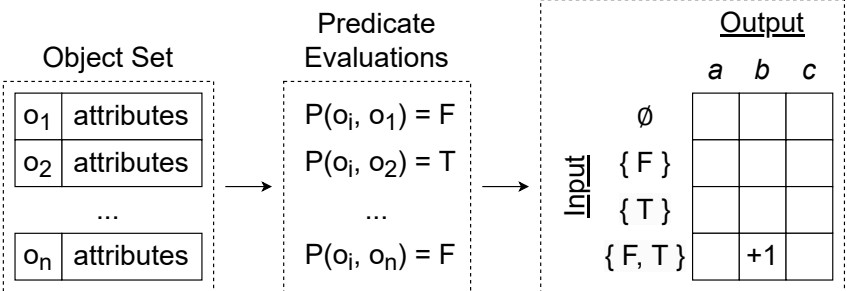

Figure 4: Illustration of a relation group containing predicate $P$ being evaluated with respect to some object $o_i$. The value of $P$ is computed for each pair $(o_i, o_j)$ and the values (T, F) that appeared are tracked. In this case, both values (T and F) are present, so we increment the table under the corresponding output for this observation. In this example, the output is $b$; this is determined by a separate process.

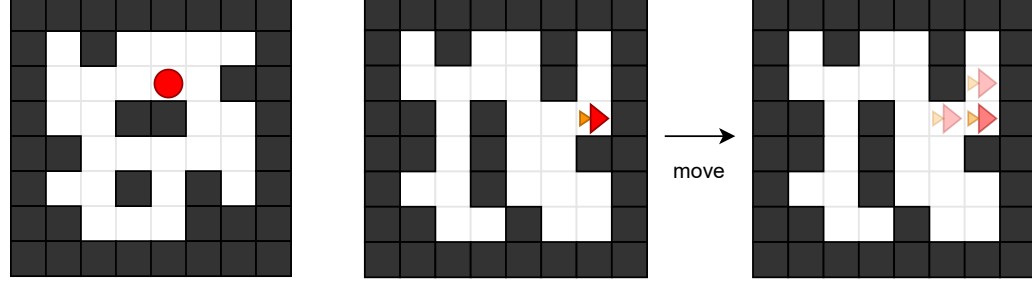

(a) example from the walls domain; player is denoted by a red circle

(b) transition in the fish domain showing a distribution over future states; the fish is denoted by colored triangles

Figure 5: Examples of two domains: walls and fish

