# OpenReview forum: "QORA: Zero-Shot Transfer via Interpretable Object-Relational Model Learning"
_ICLR.cc/2024/Conference — Submitted to ICLR 2024_

### Official Review · Reviewer_XA1i · 2023-10-21

**Soundness:** 2 fair
**Presentation:** 2 fair
**Contribution:** 2 fair
**Rating:** 3
**Confidence:** 2

**Summary:**

The paper proposes a QORA – algorithm that constructs versatile models from domain-agnostic object-based state representations, addressing the generalization challenges faced by current approaches. On a proposed benchmark to depict the generalization capability, they depict better predictive accuracy with significantly fewer observations compared to their baselines, showcasing zero-shot transfer to altered environments, and quick adaptation capabilities to tasks with previously unseen object interactions.

**Strengths:**

The presented benchmarking environment in this study is thoughtfully designed, offering a clear understanding of the rationale behind each component, including walls, doors, and the fish system.
Additionally, Section 3, which delineates the different elements of the proposed approach, provides readers with a comprehensive insight into the methodology's operation.

**Weaknesses:**

The proposed benchmark exhibits limitations in terms of its representation of real-world complexities. Despite featuring three distinct testing environments, it falls short of capturing the nuanced and open-ended nature of real-world concepts. The grid world benchmark, while valuable for assessing agent generalization capabilities, may be considered relatively straightforward and may not fully address the complexities exhibited by current deep learning systems with intricate behaviors. Moreover, its discrete nature may not adequately mirror the continuous nature of many real-world generalization challenges.
While the paper attempts to evaluate against neural network-based approaches, it predominantly relies on relatively simplistic methods, with the best-performing one being the CNN. I recommend the authors expand the range of neural network baselines for a more comprehensive comparison with the proposed approach. Additionally, extending the evaluations to larger grid sizes would provide valuable insights into how performance scales.

**Questions:**

The paper could benefit from providing more detailed information on hyperparameters and implementation specifics. Including a section on the potential broader impact of the proposed approach would also enhance the reader's understanding and appreciation of the work.
As mentioned in Weaknesses, I would recommend the authors evaluate against a more exhaustive set of approaches.

---

> ### Author Response · Authors · 2023-11-23
> **Response (1/2)**
>
> > [1] The proposed benchmark exhibits limitations in terms of its
>     representation of real-world complexities. Despite featuring
>     three distinct testing environments, it falls short of capturing
>     the nuanced and open-ended nature of real-world concepts.
>
> When assessing the capabilities of QORA, it is important
> to compare it to similar work to ensure that the comparison is fair.
> While deep reinforcement learning *in general* is applied to a
> wide variety of problems, if we consider the specific subtask we are
> focusing on here -- object-oriented transition modeling without
> domain knowledge, with a focus on generalization, efficiency, and
> interpretability -- we see that existing methods struggle in many
> ways, while QORA performs enormously better. For example, transition
> modeling is a notoriously difficult problem, resulting in modern deep
> reinforcement learning approaches often being model-free. Even notable
> model-based techniques, such as MuZero, do not actually attempt to
> model the environment's full transition function (and even then, the fact that
> MuZero makes model-based reinforcement learning work is a big
> achievement). When compared with non-deep methods, QORA is applicable
> to *substantially* more complex domains, as it learns rules in a
> general form and requires no domain-specific information. We have
> revised the introduction to highlight some of these points and
> address possible confusion.
>
> > [2] The grid world benchmark, while valuable for assessing agent
>     generalization capabilities, may be considered relatively
>     straightforward and may not fully address the complexities
>     exhibited by current deep learning systems with intricate
>     behaviors.
>
> We agree that these test domains are valuable for
> assessing generalization, as this is part of the motivation for each
> environment's design. However, we believe that our experiments
> (especially those in the revised paper) demonstrate that our
> environments, though they may *seem* straightforward to humans,
> are actually far from it: as shown in Figures 2a and 2b of the
> revision, the deep-learning baselines are unable to learn even our
> simplest domain *and* they show a massive increase in error
> after transfer. In the same task, QORA demonstrates perfect zero-shot
> transfer to an even more complex setting after training on far less
> data.
>
> > [3] Moreover, its discrete nature may not adequately mirror the
>     continuous nature of many real-world generalization challenges.
>
> The continuous nature of the world does indeed bring many
> challenges, which we should be able to tackle in future work based
> the ideas proposed in the current paper. However, as QORA is an
> initial prototype in this research direction, dealing with these
> generalizations is beyond the scope of this paper.
>
> > [4] While the paper attempts to evaluate against neural
>     network-based approaches, it predominantly relies on relatively
>     simplistic methods, with the best-performing one being the CNN. I
>     recommend the authors expand the range of neural network
>     baselines for a more comprehensive comparison with the proposed
>     approach.
>
> Unfortunately, there are not many existing approaches that
> are applicable to the specific problem we are concerned with; we have
> revised the introduction to be more clear about this. We have also
> included an additional baseline method, MHDPA, though it performs
> quite similarly to NPE no matter how we adjust its parameters or
> architecture. We have also added details to the new appendix to
> clarify that the implementations of these two baselines are quite
> sophisticated (especially relative to the problem they are being
> applied to).
>
> We have also removed comparisons to the CNN, as it was a hand-crafted
>     architecture specifically designed to solve the walls domain.
>     Thus, it incorporated domain knowledge, which disqualifies it
>     from being a valid approach within the context of the paper.

---

> ### Author Response · Authors · 2023-11-23
> **Response (2/2)**
>
> > [5] Additionally, extending the evaluations to larger grid sizes
>     would provide valuable insights into how performance scales.
>
> The revision contains new zero-shot transfer experiments
> that go from 8x8 to 64x64 and 128x128. We also demonstrate its
> capabilities in going from a small number of bulbs to a large number. In
> either case, once QORA learns the proper relational rules, the
> world's size has no bearing on its predictive accuracy, so the domain
> could be scaled up to any size without affecting QORA's accuracy.
>
> In terms of computational performance, as with all existing work,
>  QORA's runtime (for both training and prediction) is related to the
>  number of objects; thus, scaling up the world does increase the
>  training and evaluation cost. This actually makes QORA's
>  generalization ability even more impressive -- it can be trained on
>  small worlds at low cost and immediately deployed in larger ones
>  without having to invest the extra cost to learn the bigger world
>  from scratch. In the new Figure 3c of the revision, we demonstrate
>  the utility of this approach.
>
> > [6] The paper could benefit from providing more detailed
>     information on hyperparameters and implementation specifics.
>
> QORA has only a single hyperparameter, $\alpha$, which is
> discussed in Section 3 and whose value is included in each relevant
> experiment. In the revision, we have outlined the design/training of the neural-network baselines
> used in our experiments in the new Appendix.
>
> > [7] Including a section on the potential broader impact of the
>     proposed approach would also enhance the reader's understanding
>     and appreciation of the work.
>
> We have revised the introduction to be more explicit about
> the motivation behind our approach and its potential for paving the
> way towards more sophisticated future applications.
>
> > [8] As mentioned in Weaknesses, I would recommend the authors
>     evaluate against a more exhaustive set of approaches.
>
> Please see our response to weakness \#4

---

### Official Review · Reviewer_95NR · 2023-10-31

**Soundness:** 3 good
**Presentation:** 2 fair
**Contribution:** 2 fair
**Rating:** 5
**Confidence:** 3

**Summary:**

In their submission, the authors present QORA, a novel algorithm that seeks to address the persistent challenge of generalization in reinforcement learning. QORA distinguishes itself by utilizing an object-oriented framework capable of forming models from a domain-agnostic, object-based state representation. This approach allows it to effectively handle environments with stochastic transitions.

Key contributions of the paper include:

1. Algorithmic Development: The introduction of QORA, which can efficiently construct expressive models that are demonstrated to solve a diverse array of domains.
2. Generalization Capability: QORA's design enables it to achieve perfect predictive accuracy in the authors' test domains, far surpassing the baseline neural network model in terms of the number of observations needed by nearly four orders of magnitude.
3. Zero-Shot Transfer: The algorithm's capacity for zero-shot transfer is particularly noteworthy, as it can adapt to environments that have been modified without the necessity for retraining.
4. Adaptability: The adaptability of QORA is further evidenced by its rapid learning curve when faced with tasks that include interactions with new objects not present in the training set.
5. Interpretability of Results: QORA's ability to generate easily interpretable rules is a significant step towards bridging the gap between performance and understandability in machine learning models.
6. Benchmark Suite: The authors also contribute a novel benchmark suite tailored to assess the generalization capabilities of learning algorithms, which is a valuable asset for future research in the field.

Overall, QORA represents an advancement in the pursuit of generalizable reinforcement learning algorithms, with a clear emphasis on efficiency, transferability, and interpretability.

**Strengths:**

Originality:

- QORA's approach to utilizing object-oriented representations for state transitions in RL is a refreshing direction that diverges from standard neural network methodologies. It contributes to the field by removing the limitations seen in prior model-based RL methods.
- The paper's novelty is also evident in the creative combination of interpretability and generalization, which are often challenging to achieve simultaneously in RL.

Quality:
- The authors make a good effort to provide benchmarks that can directly evaluate the particular angles they investigate.

Clarity:

- The interpretability of QORA's learned rules is a testament to the clarity of the approach, which is commendably communicated through examples in the paper.

Significance:

- The significant reduction in sample complexity and successful zero-shot transfer capability indicate that QORA could have a substantial impact on the efficiency of RL models.

**Weaknesses:**

1. Overall, the writing is overly complex, detracting from the key methods involved.
2. The proposed model is difficult to conceptualize, especially without a supporting figure.
3. The proposed benchmarks, while suitable as toy tests, do not extend to nuanced representation learning and remain within grid worlds, where performance does not always translate to more realistic settings.
4. There are technical imprecisions; contrary to the authors' claims, CNNs can support variable length inputs, and architectures using 3x3 convolutions can model longer-term dependencies through mechanisms like average pooling, self-attention, max pooling, and CNN cascades.
5. The neural network details used for comparison are not provided, which is a critical omission for reproducibility and transparency.
6. Figures intended for side-by-side comparison have differing scales, which could mislead the interpretation of the results.
7. The evaluation is not sufficiently robust and lacks the depth needed to substantiate the claims made.

**Questions:**

- **Writing Clarity:**
The manuscript could benefit from a more streamlined exposition. Is a revision feasible to improve clarity and conciseness, particularly in the methodological description of QORA?

- **Model Visualization:**
Including a figure to visualize QORA's architecture may aid in understanding. Could such a figure be provided?

- **Benchmark Scope:**
The benchmarks focus on grid-world environments. Can you extend these to more complex settings to better illustrate QORA's generalization?

- **Technical Precision:**
Clarification is needed on the statements about CNNs' capabilities. Can you reconcile these with the known utility of CNNs in handling variable input lengths and context-length dependencies?

- **Baseline Details:**
A detailed description of the neural network baselines would be beneficial. Could you provide this additional context?

- **Figure Consistency:**
The varying scales in comparative figures may lead to misinterpretation. Can you ensure uniform scales across all relevant figures for clarity?


Addressing these points could significantly improve the quality and impact of the work, and would improve substantially any conclusions drawn.

---

> ### Author Response · Authors · 2023-11-23
> **Response (1/2)**
>
> > [W1] Overall, the writing is overly complex, detracting from the key methods involved.
>
> Some parts of the paper may be technical, but QORA is
>  quite complex by itself and unfortunately the limited space of ICLR
>  does not allow for a more detailed explanation.
>
> > [W2] The proposed model is difficult to conceptualize, especially
>     without a supporting figure.
>
> We have added a figure to the Appendix, with a reference in Section 3.2, to help the reader
> understand the overall operation of our algorithm.
>
> > [W3] The proposed benchmarks, while suitable as toy tests, do not
>     extend to nuanced representation learning and remain within grid
>     worlds, where performance does not always translate to more
>     realistic settings.
>
> QORA is not limited to grid worlds; instead, it can
> operate in a wide variety of scenarios that follow causality
> principles that can be conceptualized by predicate logic. Hence, the
> environments included in the paper should be viewed as examples that
> demonstrate specific aspects of QORA's functionality rather than its
> inherent limitations. To highlight this further, the revised paper
> includes experiments with a new domain where QORA has to learn the
> operation of a switch that toggles a number of light bulbs. As given
> in the new conclusion, future work will extend QORA to additional
> modes of operation (e.g., planning, exploration) and explore
> domains with more complex object interactions. We consider these
> topics beyond the scope of the current paper.
>
> > [W4] There are technical imprecisions; contrary to the authors' claims, CNNs can support variable length inputs, and architectures using 3x3 convolutions can model longer-term dependencies through mechanisms like average pooling, self-attention, max pooling, and CNN cascades.
>
> Indeed, we could've worded this better. The intent was to communicate that CNNs would not work (at least, not in a reasonable way) with our object-based state representation. In the revision, the relevant text has been removed due to restructuring, so the problem is no longer present.
>
> > [W5] The neural network details used for comparison are not provided, which is a critical omission for reproducibility and transparency.
>
> This information eventually could be obtained
> from the release of our neural-network test scripts, along with a
> QORA implementation. But in the mean time, the revision contains an
> Appendix with information on the neural networks used in the
> comparison.
>
> > [W6] Figures intended for side-by-side comparison have differing
>     scales, which could mislead the interpretation of the results.
>
> It is unclear which plots the reviewer refers to and/or which
> axes. If this is the y-axis in the six graphs of Figure 2, there is
> absolutely no way that error curves that peak at 0.4 can be on the
> same scale as those that reach 100. This would make the former
> basically a flat line at zero. Instead, the figures are scaled to
> make the various curves clearly visible. If the question refers to
> Figure 3 or 4, the various plots therein are either on the same scale
> or they show completely different metrics that are not meant to be
> compared (e.g., space complexity of QORA vs error).
>
> > [W7] The evaluation is not sufficiently robust and lacks the depth needed to substantiate the claims made.
>
> We believe that our experiments are quite thorough,
> especially with the new results included in the revision (e.g.,
> zero-shot transfer to 128x128 worlds, the light bulb domain). If the
> reviewer has the specifics about which parts of the evaluation are
> not robust and/or which claims are unsubstantiated, we would be
> interested in addressing them in the camera-ready version.

---

> ### Author Response · Authors · 2023-11-23
> **Response (2/2)**
>
> > [Q1] Writing Clarity: The manuscript could benefit from a more
>     streamlined exposition. Is a revision feasible to improve clarity
>     and conciseness, particularly in the methodological description
>     of QORA?
>
> Due to the nature of the content, as well as the
> need to include details for future readers, it is difficult to make
> Section 3 any shorter. It is also unlikely that making a description
> shorter can make it more clear. In any case, to help ease the reader
> through the description of our algorithm, we have added more
> explanations to parts of QORA, especially in Section 3.2, and
> introduced Figure 4 in the revision.
>
> > [Q2] Model Visualization: Including a figure to visualize QORA's architecture may aid in understanding. Could such a figure be provided?
>
> Yes, Figure 4 in the appendix visualizes some of the structures described in Section 3.2.
>
> > [Q3] Benchmark Scope: The benchmarks focus on grid-world
>     environments. Can you extend these to more complex settings to
>     better illustrate QORA's generalization?
>
> The fact that the included environments are primarily
> grid-worlds is not due to a limitation of QORA; in fact, the revision
> includes a new light-bulb environment that is not a grid-world (it
> actually has no specific positional information at all). Rather,
> grid-worlds are a convenient tool for helping the reader understand
> the workings of the environment. QORA can learn rules in a fairly
> general form, but many of these domains would make little sense to
> humans (e.g., relations in a 7-dimensional space) or be unnecessarily
> complex (e.g., rules with hundreds of predicates). Instead, the paper
> urges the reader to focus on the essence of QORA's generalization
> capability: QORA learns rules *as simple as possible* to better
> reflect the underlying transition dynamics of the environment
> *without* relying on information specific to some arbitrary
> subset of states. This property holds no matter what environment QORA
> is applied to, while the same universally does *not* hold for
> prior work (as demonstrated by Figure 2b in the revised paper).
>
> > [Q4] Technical Precision: Clarification is needed on the statements
>     about CNNs' capabilities. Can you reconcile these with the known
>     utility of CNNs in handling variable input lengths and
>     context-length dependencies?
>
> Please see response to weakness #4 (W4).
>
> > [Q5] Baseline Details: A detailed description of the neural network baselines would be beneficial. Could you provide this additional context?
>
> Please see response to weakness #5 (W5).
>
> > [Q6] Figure Consistency: The varying scales in comparative figures
>     may lead to misinterpretation. Can you ensure uniform scales
>     across all relevant figures for clarity?
>
> Please see response to weakness #6 (W6).

---

### Official Review · Reviewer_Kowy · 2023-10-31

**Soundness:** 3 good
**Presentation:** 2 fair
**Contribution:** 2 fair
**Rating:** 3
**Confidence:** 4

**Summary:**

This paper proposes a new object-oriented transition model, Quantified Object Relation Aggregator (QORA). Different from the previous work, DOORMAX, it can learn the general class of transition rules, which are interpretable and cover stochastic transitions. Additionally, they also proposed a new benchmark to evaluate the object-oriented transition models. In this environment, there are several object classes which attributes and transition rules are different, and one of the class has a stochasticity in its movement. In this paper, the authors evaluated their proposed model, QORA outperformed the baselines including several Neural Network models, and it showed a good generalization performance for diverse size of rooms, and can be trained for the stochastic transitions also.

**Strengths:**

- This paper proposes a new object-oriented transition model which can cover general rules of stochastic transitions.
- To evaluate the object-oriented transition model, a new benchmark is proposed, and it is reasonably designed to evaluate the object-oriented models.
- The evaluation is done step-by-step, which can make the reader understand the strengths of their model easily.
- Unseen size of environments, and object attributes are evaluated for showing the generalization performance of their model.

**Weaknesses:**

- Their presentation looks not ready to be published yet in many aspects.
    - Ambiguous expression or not introduced terms.
        - In the introduction, the terms "generalization" (the model's ability to make effective decisions when exposed to novel inputs), "interpretability" (how easily its learned parameters can be inspected and understood by a human), and "robustness" (the predictability of its behavior on arbitrary inputs) need clarification. Explain the distinction between "novel" inputs (previously unseen) and "arbitrary" inputs (inputs that may not follow a specific pattern or structure) to avoid confusion.
        - On page 4, when discussing the player's ability to swap its color with the "new change-color action," it's important to introduce the "new change-color action" before using it in the sentence. Provide a brief explanation or definition of this action to ensure readers understand its context and purpose.
        - On page 5, when referring to the "best" candidate, explicitly define what "best" means within the context of the paper. Clarify whether "best" refers to candidates that are most relevant for prediction, those with the highest confidence scores, or some other criteria.
        - Regarding the explanation of boosting and the working set in the sentence on page 5, consider whether this explanation is necessary. If it adds value to the reader's understanding of how the working set is updated, provide a concise summary of how boosting is related to the working set, ensuring that it enhances clarity.
        - In section 3.2, provide a clear definition or explanation of the "learnable module" within the context of Quantified Function Learning. Specify its purpose and functionality to ensure the reader understands its role in the paper.
        - Equation 2 needs clarification regarding the conditions for movement to the right and the meaning of first and second coordinates.
        - The explanation of the relation group is insufficient, and details about its design or training architecture are lacking.
        - The meaning of "c" is not provided in Section 3.3.
        - In Section 4, different notations for different meanings should be used to avoid reader confusion.
        - The abbreviation "EMD" is used before it is introduced.
    - The analysis of experimental results needs improvement
        - The use of $m=1$ for Neural Networks and larger $m$ for DOORMAX and QORA should be justified more, as it may raise fairness concerns.
        - More analysis is needed to explain the errors that occur when $m=1000" for QORA, beyond the fact that a single layout is used.
        - The explanation for DOORMAX's inability to resolve the effect of the change-color action is insufficient.
    - There are several literatures studied the object-centric or object-oriented representation for the Reinforcement Learning tasks, but they are missed. For example, in [1], the unsupervised object-centric representation for model-free reinforcement learning agent is investigated, and in [2], the world model learning of given object attributes is studied.
    - The evaluation is limited to a synthetic environment, and the paper should discuss the potential for extending the model to more realistic environments and the expected limitations in such cases.
    - The choice of baselines lacks diversity, and the paper could benefit from considering relevant studies in the object-centric representation field [3,4,5].

[1] Yoon, Jaesik, et al. "An investigation into pre-training object-centric representations for reinforcement learning." arXiv preprint arXiv:2302.04419 (2023).

[2] Sancaktar, Cansu, Sebastian Blaes, and Georg Martius. "Curious exploration via structured world models yields zero-shot object manipulation." Advances in Neural Information Processing Systems 35 (2022): 24170-24183.

[3] Locatello, Francesco, et al. "Object-centric learning with slot attention." Advances in Neural Information Processing Systems 33 (2020): 11525-11538.

[4] Singh, Gautam, Yeongbin Kim, and Sungjin Ahn. "Neural systematic binder." The Eleventh International Conference on Learning Representations. 2022.

[5] Jiang, Jindong, et al. "Object-centric slot diffusion." arXiv preprint arXiv:2303.10834 (2023).

**Questions:**

All questions are addressed in the "Weaknesses" section.

### Additional Comments
In general, the paper's presentation requires significant improvement before publication. Additional references could provide a more comprehensive context for the work. The writing and analysis for the experiments should be further developed, including more detailed explanations of the model.

---

> ### Author Response · Authors · 2023-11-23
> **Response (1/3)**
>
> > [1A] In the introduction, the terms "generalization" (the
>     model's ability to make effective decisions when exposed to
>     novel inputs), "interpretability" (how easily its learned
>     parameters can be inspected and understood by a human), and
>     "robustness" (the predictability of its behavior on arbitrary
>     inputs) need clarification. Explain the distinction between
>     "novel" inputs (previously unseen) and "arbitrary" inputs
>     (inputs that may not follow a specific pattern or structure)
>     to avoid confusion.
>
> The revised introduction no longer uses terms "novel
> input" or "arbitrary input."
>
> > [1B] On page 4, when discussing the player's ability to swap its
>     color with the "new change-color action," it's important to
>     introduce the "new change-color action" before using it in
>     the sentence. Provide a brief explanation or definition of
>     this action to ensure readers understand its context and
>     purpose.
>
> It is a bit unclear what the reviewer means because
> "changing color" is not an ambiguous action. From the rest of the
> paragraph in the paper, it should be clear that a) colors are
> binary; b) players can flip their color to the opposite using
> this action; and c) the purpose of changing the color is to be
> able to pass through doors that have the same color. In any case,
> we have rephrased this sentence to flow better, but feel that the
> rest of the paragraph is pretty self-explanatory.
>
> > [1C] On page 5, when referring to the "best" candidate,
>     explicitly define what "best" means within the context of the
>     paper. Clarify whether "best" refers to candidates that are
>     most relevant for prediction, those with the highest
>     confidence scores, or some other criteria.
>
> Sentence rephrased.
>
> > [1D] Regarding the explanation of boosting and the working set
>     in the sentence on page 5, consider whether this explanation
>     is necessary. If it adds value to the reader's understanding
>     of how the working set is updated, provide a concise summary
>     of how boosting is related to the working set, ensuring that
>     it enhances clarity.
>
> Yes, boosting is an essential part of QORA, which is a
> method for iteratively improving a model by focusing on its
> current weaknesses. The paper explains this in fairly unambiguous
> terms: "candidates in the working set do not receive
> observations that the best hypothesis predicts with high
> confidence, thereby magnifying the score of candidates that are
> well-suited to correct existing errors." Due to limited space and
> the fact that boosting is a well-known concept in the field of
> ML, we do not believe that elaborating on the rationale for using
> this method is warranted. Instead, the revision now cites the
> paper that invented boosting, while as a brief reference, the reviewer is directed to
> the following page for more information:
>
> https://en.wikipedia.org/wiki/Boosting_(machine_learning)
>
> > [1E] In section 3.2, provide a clear definition or explanation
>     of the "learnable module" within the context of Quantified
>     Function Learning. Specify its purpose and functionality to
>     ensure the reader understands its role in the paper.
>
> The paper does not contain the phrase "learnable module", which makes it difficult for
> us to specify its purpose and functionality in the context of
> QORA. If the reviewer uses this term to refer to deep learning,
> where modules are composed to produce complex neural-network
> architectures, this concept is not relevant to the current paper.
> Instead, we are proposing an entirely new (non-neural-network
> based) algorithm for learning predictive models, where QORA (and,
> more specifically, the topics covered in the Quantified Function
> Learning subsection) are not meant to be used as composable
> modules in this sense.
>
> > [1F] Equation 2 needs clarification regarding the conditions for
>     movement to the right and the meaning of first and second
>     coordinates.
>
> It is a bit unclear what the reviewer means, but
> generally speaking most fields agree on a convention that
> a position in 2D space is commonly described by a tuple in which
> the $x$ coordinate appears first and the $y$ coordinate appears
> second. Hence, movement to the right corresponds to an increase
> in the $x$ coordinate, which the paper confirms just above (2) by
> stating "a player $x$ can move to the right: $P(x): \exists y \in
> walls: y.pos - x.pos = (1, 0)$". In more detail, this predicate is
> true when there is a wall on the right side of the player's
> current position; hence, the move is successful iff the predicate
> is false. More broadly, the operation of QORA does not depend on
> the direction in which left/right/up/down actions move the
> player. Its algorithms would work exactly the same if the domain
> were flipped sideways, e.g., moving up would cause the $x$
> coordinate to reduce.
>
> Since the original formulas used $x$, $y$, $z$ to prepresent objects,
> which may be causing confusion with coordinates, we have changed
> these variables to $o_1$, $o_2$, $o_3$ in the revision.

---

> > ### Comment · Reviewer_Kowy · 2023-11-23
> > **Response to the author's reply**
> >
> > Thank you for your reply.
> >
> > [1B] On page 4, when discussing the player's ability to swap its color with the "new change-color action," it's important to introduce the "new change-color action" before using it in the sentence. Provide a brief explanation or definition of this action to ensure readers understand its context and purpose.
> > It is a bit unclear what the reviewer means because "changing color" is not an ambiguous action. From the rest of the paragraph in the paper, it should be clear that a) colors are binary; b) players can flip their color to the opposite using this action; and c) the purpose of changing the color is to be able to pass through doors that have the same color. In any case, we have rephrased this sentence to flow better, but feel that the rest of the paragraph is pretty self-explanatory.
> >
> > Yes, it is understandable, but what I pointed out is when using a new word (e.g., new change-color action), giving more kind introduction should be helpful (what is "new" change-color action? I checked again, I think you aimed to say this action is an introduced action for door, maybe it is better to say ..... with a new action called as change-color.
> >
> >  [1D] Regarding the explanation of boosting and the working set in the sentence on page 5, consider whether this explanation is necessary. If it adds value to the reader's understanding of how the working set is updated, provide a concise summary of how boosting is related to the working set, ensuring that it enhances clarity.
> > Yes, boosting is an essential part of QORA, which is a method for iteratively improving a model by focusing on its current weaknesses. The paper explains this in fairly unambiguous terms: "candidates in the working set do not receive observations that the best hypothesis predicts with high confidence, thereby magnifying the score of candidates that are well-suited to correct existing errors." Due to limited space and the fact that boosting is a well-known concept in the field of ML, we do not believe that elaborating on the rationale for using this method is warranted. Instead, the revision now cites the paper that invented boosting, while as a brief reference, the reviewer is directed to the following page for more information:
> >
> > https://en.wikipedia.org/wiki/Boosting_(machine_learning)
> >
> > It is okay, I didn't understand what it means, when revisiting I can understand.
> >
> > [1E] In section 3.2, provide a clear definition or explanation of the "learnable module" within the context of Quantified Function Learning. Specify its purpose and functionality to ensure the reader understands its role in the paper.
> > The paper does not contain the phrase "learnable module", which makes it difficult for us to specify its purpose and functionality in the context of QORA. If the reviewer uses this term to refer to deep learning, where modules are composed to produce complex neural-network architectures, this concept is not relevant to the current paper. Instead, we are proposing an entirely new (non-neural-network based) algorithm for learning predictive models, where QORA (and, more specifically, the topics covered in the Quantified Function Learning subsection) are not meant to be used as composable modules in this sense.
> >
> > the learnable module means, then what is learned? which parameters are learned? you commented the function is learned, then which components in the function are learned?
> >
> > [1F] Equation 2 needs clarification regarding the conditions for movement to the right and the meaning of first and second coordinates.
> > It is a bit unclear what the reviewer means, but generally speaking most fields agree on a convention that a position in 2D space is commonly described by a tuple in which the  coordinate appears first and the  coordinate appears second. Hence, movement to the right corresponds to an increase in the  coordinate, which the paper confirms just above (2) by stating "a player  can move to the right: ". In more detail, this predicate is true when there is a wall on the right side of the player's current position; hence, the move is successful iff the predicate is false. More broadly, the operation of QORA does not depend on the direction in which left/right/up/down actions move the player. Its algorithms would work exactly the same if the domain were flipped sideways, e.g., moving up would cause the  coordinate to reduce.
> >
> > Since the original formulas used , ,  to prepresent objects, which may be causing confusion with coordinates, we have changed these variables to , ,  in the revision.
> >
> > Same to the above. Yes, usually x-axis is first and y-axis is second. But you didn't notice it clearly. Kind introduction should improve the readability of your paper, so I recommended.

---

> ### Author Response · Authors · 2023-11-23
> **Response (2/3)**
>
> > [1G] The explanation of the relation group is insufficient, and details about its design or training architecture are lacking.
>
> Indeed, some of the notation used in this section is
> fairly technical, but it is based on standard first-order logic. The
> limited space in the paper does not permit going into much depth
> on these topics. In brief, relation groups are a part of the
> structure that QORA uses to represent rules. In particular, they
> enable QORA to efficiently keep track of quantified groups of
> predicates and generate expressive outputs. They are not
> trainable, so there is no "training architecture" related to
> them. Instead, QORA constructs more complex relation groups
> through its iterative process described in Section 3.
>
> > [1H] The meaning of "c" is not provided in Section 3.3.
>
> It appears the reviewer missed that $(c, m, a)$ triplets
> were previously introduced at the beginning of Section 3, where
> the paper states "We separate transition rules based on the
> (class, member attribute type, action) $(c, m, a)$ triplet they
> apply to and predict changes independently for each attribute of
> every object." In the revision, we have added text to Section
> 3.3 to remind the reader of this previous definition.
>
> > [1I] In Section 4, different notations for different meanings should be used to avoid reader confusion.
>
> It is unclear what notation the reviewer refers to. Any variables that were reused (e.g., $m$) were in clearly different contexts, but to ensure there could be no notation-related confusion throughout the paper, we have gone through and changed some of the variables (e.g., the $m$ used in Section 4 is now $n$).
>
> > [1J] The abbreviation "EMD" is used before it is introduced.
>
> Thank you for catching this; we have fixed it.
>
> > [2A] The use of $m=1$ for Neural Networks and larger $m$ for
>     DOORMAX and QORA should be justified more, as it may raise
>     fairness concerns.
>
> It appears the reviewer got this backwards -- as
> mentioned in Section 4.1 "neural networks used $m=1$ to increase
> learning speed." Additionally, as shown in the original Figure
> 2(c), decreasing $m$ leads to better performance, which means
> that neural networks obtain the maximum advantage over QORA when
> running with $m=1$. This happens because lower $m$ means the
> algorithm receives more variety in the training data. The correct
> interpretation of the results in Section 4 should be: even
> though neural networks run with the best possible $m$, they
> still cannot come anywhere near QORA's efficiency and ability to
> learn. To prevent this confusion in the future, we have added a
> new Figure 2(a) in the revision that tests the neural network
> baselines with varied episode lengths (now called $n$ instead of
> $m$). This new plot perhaps makes it more clear that having
> $m=1$, i.e., generating a new level for every observation, leads
> to best performance.
>
> > [2B] More analysis is needed to explain the errors that occur
>     when $m=1000$ for QORA, beyond the fact that a single layout
>     is used.
>
> We appreciate the comment, but disagree that more
> analysis is needed to explain the poor performance with larger
> $m$. This follows from common sense. When a model is trained on
> only a single level, it will still *typically* learn most of
> the information it needs, but it's possible for a
> randomly-generated level to be unusually difficult (e.g., to trap
> the player in a series of long hallways, not allowing it to
> explore the entire level as freely as normal). If the agent is
> "unlucky" enough to have this as its only experience, it may
> not be able to gather enough evidence over the course of 1,000
> random actions to properly learn the environment's general rules.
> This is a problem that is not unique to QORA; if an agent is not
> in control of the data it receives, learning may become
> arbitrarily difficult.
>
> > [2C] The explanation for DOORMAX's inability to resolve the
>     effect of the change-color action is insufficient.
>
> Although the revision no longer compares against
> DOORMAX or points out its inability to work with the doors
> domain, we can elaborate on this topic in the response. DOORMAX
> can learn effects of two types: arithmetic (add or subtract a
> constant) and assignment (set to a constant value). As part of
> the learning process, it attempts to determine which of these two
> types of effects is actually being enacted by a given rule.
> Notably, for a given transition, DOORMAX will not make *any*
> prediction if there is still a rule that has ambiguous effects.
> In the case of the doors domain, the effect toggles an attribute
> from $0 \to 1$ or $1 \to 0$, which could be considered either
> arithmetic *or* assignment (since the initial value of the
> attribute is given as input to the rule). Thus, DOORMAX is unable
> to disambiguate the effect type, which forces it to give up and
> make no prediction.

---

> ### Author Response · Authors · 2023-11-23
> **Response (3/3)**
>
> > [3] There are several literatures studied the object-centric or
>     object-oriented representation for the Reinforcement Learning
>     tasks, but they are missed. For example, in [1], the unsupervised
>     object-centric representation for model-free reinforcement
>     learning agent is investigated, and in [2], the world model
>     learning of given object attributes is studied.
>
> We do not believe "missed" is the right word, as many of
> these works are not relevant to our particular problem (transition
> modeling). For example, [1] seems to deal with the problem of
> perception (i.e., converting visual input to an object-oriented
> representation), while in the current paper, we assume such a system
> exists and work directly with objects without focusing on how they
> were processed. Paper [2] deals with object-oriented transition
> modeling, like us, but because it assumes domain-specific knowledge,
> it is not comparable to our method. Specifically, the GNN structure
> described in that paper assumes knowledge of a singular
> agent-controlled object within the environment, which is not always a
> valid assumption; for example, in our fish domain, there are multiple
> objects that are partially controlled by the agent (or, depending on
> perspective, there is *no* object that is directly controlled by
> the agent). If this domain knowledge were to be removed from the GNN,
> it would become incredibly similar to the NPE baseline that we have
> already included in the paper. However, since it was mentioned here,
> we have added a citation to [2] in our revised introduction.
>
> > [4] The evaluation is limited to a synthetic environment, and the
>     paper should discuss the potential for extending the model to
>     more realistic environments and the expected limitations in such
>     cases.
>
> Many reinforcement learning experiments are done in synthetic
> environments due to the high cost of real-world experiments
> (especially when millions of training iterations are required, as
> with most deep-learning approaches). In this case, the environments
> we introduce are designed to test specific aspects of QORA's learning
> without adding unnecessary complexity. In the revised introduction, we
> mention that future work will expand QORA's capabilities and address
> more challenging domains.
>
> > [5] The choice of baselines lacks diversity, and the paper could
>     benefit from considering relevant studies in the object-centric
>     representation field [3,4,5].
>
> We do not believe the listed papers [3, 4, 5] are relevant
> to our problem as they mostly deal with learning object-centric
> representations. Instead, the current paper assumes such a
> representation is given, but the goal is to learn object-based state
> transition models of a given world. The revised introduction should
> explain this better, in addition to pointing out that the field lacks
> baselines that fit our requirements (i.e., model-based,
> object-oriented domain learning with no a-prior knowledge). While we
> found another such technique (MHDPA) and included comparison against
> it in the revision, the outcome is not promising, i.e., it performs
> as poorly as NPE.

---

> > ### Comment · Reviewer_Kowy · 2023-11-23
> > **response to the author's reply**
> >
> > [3] There are several literatures studied the object-centric or object-oriented representation for the Reinforcement Learning tasks, but they are missed. For example, in [1], the unsupervised object-centric representation for model-free reinforcement learning agent is investigated, and in [2], the world model learning of given object attributes is studied.
> > We do not believe "missed" is the right word, as many of these works are not relevant to our particular problem (transition modeling). For example, [1] seems to deal with the problem of perception (i.e., converting visual input to an object-oriented representation), while in the current paper, we assume such a system exists and work directly with objects without focusing on how they were processed. Paper [2] deals with object-oriented transition modeling, like us, but because it assumes domain-specific knowledge, it is not comparable to our method. Specifically, the GNN structure described in that paper assumes knowledge of a singular agent-controlled object within the environment, which is not always a valid assumption; for example, in our fish domain, there are multiple objects that are partially controlled by the agent (or, depending on perspective, there is no object that is directly controlled by the agent). If this domain knowledge were to be removed from the GNN, it would become incredibly similar to the NPE baseline that we have already included in the paper. However, since it was mentioned here, we have added a citation to [2] in our revised introduction.
> >
> > [1] focused on the model-free reinforcement learning on top of the pretrained object-centric perception model.
> >
> >
> > [5] The choice of baselines lacks diversity, and the paper could benefit from considering relevant studies in the object-centric representation field [3,4,5].
> > We do not believe the listed papers [3, 4, 5] are relevant to our problem as they mostly deal with learning object-centric representations. Instead, the current paper assumes such a representation is given, but the goal is to learn object-based state transition models of a given world. The revised introduction should explain this better, in addition to pointing out that the field lacks baselines that fit our requirements (i.e., model-based, object-oriented domain learning with no a-prior knowledge). While we found another such technique (MHDPA) and included comparison against it in the revision, the outcome is not promising, i.e., it performs as poorly as NPE.
> >
> > What I mentioned is, even though there are several studies relevant to object-centric representation, and your paper is focusing on the object-oriented transition model learning, but they are not considered as relevant works or baselines. Yes, maybe they cannot be the baselines due to several reasons, then you need to discuss it as related work, and why they are not compared in your experiments I think.

---

> ### Comment · Reviewer_Kowy · 2023-11-23
> **response to the author's reply**
>
> [1G] The explanation of the relation group is insufficient, and details about its design or training architecture are lacking.
> Indeed, some of the notation used in this section is fairly technical, but it is based on standard first-order logic. The limited space in the paper does not permit going into much depth on these topics. In brief, relation groups are a part of the structure that QORA uses to represent rules. In particular, they enable QORA to efficiently keep track of quantified groups of predicates and generate expressive outputs. They are not trainable, so there is no "training architecture" related to them. Instead, QORA constructs more complex relation groups through its iterative process described in Section 3.
>
> Thank you for your kind explanation. Could you update the details in your appendix? It could be very helpful to understand your paper more.
>
> [1I] In Section 4, different notations for different meanings should be used to avoid reader confusion.
> It is unclear what notation the reviewer refers to. Any variables that were reused (e.g., ) were in clearly different contexts, but to ensure there could be no notation-related confusion throughout the paper, we have gone through and changed some of the variables (e.g., the  used in Section 4 is now ).
>
> each starting state m in section 4 and m in "the (class, member attribute type, action) (c, m, a) triplet they apply to and predict changes independently for each attribute of every object" are same?
>
>
> I cannot check every reply in the discussion period due to the limitation of the time, but I will check more even after the discussion period.
>
> Thank you for your reply.

---

### Official Review · Reviewer_RNSv · 2023-11-01

**Soundness:** 3 good
**Presentation:** 3 good
**Contribution:** 3 good
**Rating:** 6
**Confidence:** 2

**Summary:**

This paper introduces QORA, a new algorithm for learning interpretable object-relational models that can efficiently solve reinforcement learning tasks with zero-shot generalization. QORA represents states as sets of objects and attributes. It constructs transition rules for each object type by iteratively generating relational predicate hypotheses and combining them using first-order logic with quantification. In experiments on three environments, QORA achieves zero error with orders of magnitude fewer observations than neural networks. It demonstrates strong generalization via zero-shot transfer and rapid adaptation to new object types and interactions. The learned conditional probability rules are compact and interpretable. Overall, QORA advances object-oriented RL by increasing applicability to complex stochastic environments while retaining interpretability.

**Strengths:**

- This paper presents a novel object-oriented RL algorithm with strong empirical results on efficiency, generalization, and interpretability. Compared to DOORMAX, QORA is able to solve more complex environments (e.g., doors) and perform zero-shot transfer to modified environments. Moreover, the rules QORA deduced are also interpretable.
-  The source code of both QORA’s reference implementation and the benchmark suite will be public, which is beneficial to the community.

**Weaknesses:**

- While QORA tacles more challenging environments than prior works (e.g., DOORMAX_D in Marom and Rosman, 2018), the evaluations are still on small-scale games, which limits its wide applicability.
- There are no theoretical guarantees provided for convergence or sample complexity.

This paper is not at all in my area. I am not familiar with OO-MDPs and the follow up works. I think the results presented in this paper is promising compared to existing works but not sure whether it makes adequte contribution in this field.

**Questions:**

N/A

---

> ### Author Response · Authors · 2023-11-23
>
> > While QORA tacles more challenging  environments than prior works (e.g., DOORMAX\_D in Marom and Rosman, 2018), the evaluations are still on small-scale games, which limits its wide applicability.
>
> We appreciate the comment, but QORA is applicable to a
> much larger variety of environments than just those shown in the
> paper, both in terms of their size (i.e., number of objects) and
> the types of rules that govern their behavior. Furthermore, it should be
> noted that even the domains that appear simple/small require
> construction and consideration of a large number of hypotheses. For
> example, the 8x8 walls domain contains 28 external (perimeter) walls
> and 10 interior walls, whose semantics QORA isn't given (e.g., it
> has no idea that walls cannot move). Therefore, it must consider all
> possible interactions between them, as well as the position of the
> agent and action taken, to build a predictive model. Given the
> limited space in the paper, we focus on the most basic examples to
> demonstrate QORA's particular capabilities rather than achieve an
> exhaustive coverage of all possible worlds it can operate in. Future
> work will focus on extending QORA to handle game playing, exploration, and planning, but
> these topics are clearly beyond the scope of the current paper.
>
> With this in mind, the revised paper contains new experiments with
> larger (64x64 and 128x128) worlds, a new domain with light bulbs, and
> additional discussion emphasizing some of the points in the preceding
> paragraph.
>
> > There are no theoretical guarantees provided for convergence or sample complexity.
>
> Indeed, QORA tackles much more general problems than prior
> work, making theoretical analysis correspondingly more difficult and
> therefore beyond the scope of this introductory paper. However, we
> agree this is an important topic, which we plan to address in future
> publications.

---

### Author Response · Authors · 2023-11-23

We thank the reviewers for their comments. The revision contains numerous changes to address these questions, including the following major elements:

1. a new introduction, which explains that the focus of the paper is to design generalizable and interpretable algorithms that can learn the rules of fairly general object-oriented environments and create rigorous models to predict future object states without a-priori domain knowledge, which is arguably the first step towards more complex interaction with the world (e.g., exploration and planning) that we plan to examine in future work;

2. restructuring of the paper, where Section 2.1 (test domains) is moved into the experimental section to avoid creating the impression that QORA is limited to grid worlds; instead, we point out that it can work on a wide variety of domains that go well beyond those examined in the paper;

3. additional discussion and figures in Section 3 that better highlight QORA's operation;

4. a new test domain in Section 4 in which an agent operates in a room with a switch that controls a number of light bulbs;

5. new experiments that show zero-shot transfer ability of QORA to larger grids (e.g., from 8x8 to 128x128, from few bulbs to many), which also demonstrates the impressive resulting efficiency improvement (i.e., training on a small world, followed by a transfer, is much faster than direct learning on the large world);

6. removal of experimental results with CNNs, which have an unfair advantage over QORA because we programmed them with domain knowledge (i.e., the fact that only objects in the immediate vicinity of the agent affect the future state of the world), instead introducing another prior method MHDPA and showing that it performs similarly to NPE;

7. removal of neighborhood masking from NPE for similar reasons;

8. additional explanation and figures related to QORA's evaluation, as well as new experiments showing that NPE and MHDPA fail to transfer even on the simplest domain;

9. various improvements in writing and clarity; and

10. a new Appendix that documents the design, parameters, and training of the neural-network baselines (as well as details and figures relating to other parts of the paper).

---

### Meta-Review · Area_Chair_qPAw · 2023-12-04

**Metareview:**

The paper should undergo another review round. The reviewers appreciated the major rewrite (with clarity being a common thread in the comments). At the same time, issues about scalability remain, especially as the proposed benchmark was judged to be too simplistic. The focus on object-based state representation is very interesting, and I would strongly encourage the authors to resubmit their work in the new, clearer version. More complex environments would also be appreciated, to showcase the scalability properties of the method.

**Justification For Why Not Higher Score:**

Concerns on lack of scalability and clarity, leading to a major rewrite.

**Justification For Why Not Lower Score:**

N/A

---

### Decision · Program_Chairs · 2024-01-16

Reject